# Efficacy and Mechanisms of Antioxidant Compounds and Combinations Thereof against Cisplatin-Induced Hearing Loss in a Rat Model

**DOI:** 10.3390/antiox13070761

**Published:** 2024-06-24

**Authors:** Liliana Carles, Alejandro Gibaja, Verena Scheper, Juan C. Alvarado, Carlos Almodovar, Thomas Lenarz, José M. Juiz

**Affiliations:** 1Instituto de Investigación en Discapacidades Neurológicas (IDINE), School of Medicine, Universidad de Castilla-La Mancha (UCLM), Campus in Albacete, 02008 Albacete, Spain; lili_ce42@hotmail.com (L.C.); gibaja.alejandro@gmail.com (A.G.); juancarlos.alvarado@uclm.es (J.C.A.); 2Department of Otolaryngology, University Hospital “Doce de Octubre”, 28041 Madrid, Spain; carlosalmodovaralvarez@gmail.com; 3Department of Otorhinolaryngology, Head and Neck Surgery, Hannover Medical School, 30625 Hannover, Germany; scheper.verena@mh-hannover.de (V.S.); lenarz.thomas@mh-hannover.de (T.L.); 4Cluster of Excellence “Hearing4all” of the German Research Foundation, DFG, 26111 Oldenburg, Germany

**Keywords:** cisplatin, ototoxicity, oxidative stress, antioxidant

## Abstract

Cisplatin is an election chemotherapeutic agent used for many cancer treatments. Its cytotoxicity against neoplastic cells is mirrored by that taking place in healthy cells and tissues, resulting in serious adverse events. A very frequent one is ototoxicity, causing hearing loss which may permanently affect quality of life after successful oncologic treatments. Exacerbated oxidative stress is a main cytotoxic mechanism of cisplatin, including ototoxicity. Previous reports have shown antioxidant protection against cisplatin ototoxicity, but there is a lack of comparative studies on the otoprotectant activity and mechanism of antioxidant formulations. Here, we show evidence that a cocktail of vitamins A, C, and E along with Mg^++^ (ACEMg), previously shown to protect against noise-induced hearing loss, reverses auditory threshold shifts, promotes outer hair cell survival, and attenuates oxidative stress in the cochlea after cisplatin treatment, thus protecting against extreme cisplatin ototoxicity in rats. The addition of 500 mg N-acetylcysteine (NAC), which, administered individually, also shows significant attenuation of cisplatin ototoxicity, to the ACEMg formulation results in functional degradation of ACEMg otoprotection. Mg^++^ administered alone, as MgSO_4_, also prevents cisplatin ototoxicity, but in combination with 500 mg NAC, otoprotection is also greatly degraded. Increasing the dose of NAC to 1000 mg also results in dramatic loss of otoprotection activity compared with 500 mg NAC. These findings support that single antioxidants or antioxidant combinations, particularly ACEMg in this experimental series, have significant otoprotection efficacy against cisplatin ototoxicity. However, an excess of combined antioxidants and/or elevated doses, above a yet-to-be-defined “antioxidation threshold”, results in unrecoverable redox imbalance with loss of otoprotectant activity.

## 1. Introduction

Cisplatin (cis-diaminedichloroplatinum(II), CDDP) is a drug used in many cancer treatments [1]. One important adverse effect is ototoxicity [2], present in 12% to 100% of patients, depending on the dose and treatment protocol [3,4]. The resulting hearing loss is sensorineural, typically bilateral, and irreversible [5,6]. It affects mainly high frequencies (4–8 kHz in humans) [7,8,9], and lower frequencies also become affected at high doses. The prevalence of hearing loss after cisplatin chemotherapy stimulates the search for efficient treatment or prevention strategies [10,11].

One central antineoplastic mechanism of cisplatin involves oxidative stress. Upon entering cells, cisplatin forms adducts with DNA, which interferes with DNA replication and repair and in general with normal gene transcription and regulation, essential for cell survival and perpetuation [12]. Mitochondrial DNA, lacking histones, easily forms such crosslink adducts with cisplatin, which impairs essential mitochondrial functions. Importantly, gene transcription and thus protein synthesis of electron transport chain enzymes are altered, and thus their activity is affected. This compromises oxidative phosphorylation and finally energy metabolism. Defective electron transport results in oxidative imbalance with excess accumulation of incompletely reduced, highly reactive oxygen species (ROS), which readily combine with nitrogenated compounds, giving rise to reactive nitrogen species (RNS). Excess ROS/RNS and related free radicals and non-radicals lead to unchecked oxidative stress, which finally overrides the natural “antioxidant defenses” of the cell. Self-perpetuation of oxidative stress further alters gene expression due to sensitivity of nucleic acids to oxidative degradation, as well as proteins and structural lipids, with disorganization of membranes. Altered expression affects genes coding for “antioxidant” enzymes. Their activity may be also compromised by excess free radicals, further contributing to cisplatin cytotoxicity. All this finally results in the activation of cell death pathways [12,13].

The high metabolic requirements of neoplastic cells make them particularly sensitive to cisplatin cytotoxic oxidative stress. However, it also causes unwanted cytotoxicity, particularly in cells and tissues with delicate metabolic energy balance, such as the auditory receptor organ. As mentioned, cisplatin leads to ototoxic deafness in a significant number of patients [14,15]. Systemic cisplatin is transported across the blood–endolymphatic barrier and into cells in the cochlea via transport proteins including megalin, organic cation transporters, and copper transporters. Then, cisplatin–DNA adducts form in cochlear sensory and non-sensory cells [14,16]. Cisplatin accumulation is particularly intense and long-lasting in cells of the stria vascularis, which alters the endocochlear potential [17]. Regardless of cell type, structural and functional degradation of mitochondria and ROS/RNS excess, with resulting oxidative stress, are key components of cisplatin-mediated ototoxicity [18,19]. Like in tumoral cells, inactivation of enzymes involved in regulating oxidative stress also contribute to oxidative stress damage [13,20]. In addition, cisplatin-induced overexpression of NOX3 and NADPH-oxidase, virtually selective of the inner ear, makes the auditory end organ extremely prone to toxic damage by cisplatin [19,21]. Overproduction of ROS by NOX3 accelerates lipid peroxidation [14], which, among other things, increases abnormal Ca^++^ flow across outer hair cell (OHCs) membranes, which is a signal for apoptosis [22,23].

In sum, many studies support increased oxidative stress as a main cisplatin ototoxicity mechanism [13,24]. Therefore, efforts have focused on neutralizing oxidative stress damage in the cochlea with antioxidant compounds [20,25,26]. From a biomedical perspective, antioxidants are chemical species which counteract and regulate biological oxidations, keeping ROS/RNS and other reactive free radicals and non-radicals within the levels required to maintain the physiological redox state and signaling. The expectation is that in pathological conditions such as cisplatin ototoxicity, in which built-in enzymatic and non-enzymatic cell “antioxidant defenses” may be critically challenged, antioxidant supplementation may restore redox balance by scavenging free radicals or otherwise limiting the oxidative capacity of free radicals and non-radicals.

Antioxidant compounds assayed in cisplatin ototoxicity include sodium thiosulfate [27,28], D-methionine [29,30], diethyldithiocarbamate, 4-methylthiobenzoic acid, ebselen, lipoic acid [31], and N-acetylcysteine [32,33], among others [19,28,34]. Most experimental tests have been conducted in rodent models. Histological preservation of hair cells and supporting structures, threshold recovery in auditory brainstem responses (ABRs), preservation of distortion product otoacoustic emissions, and recovery of antioxidant enzyme activity or combinations thereof have been reported after antioxidant administration [13,19,28]. However, this is still not mirrored by conclusive evidence in the clinical setting [25,35], which stresses the need for more experimental, pre-clinical studies contributing further evidence on otoprotective activity and mechanisms of antioxidant therapies.

The complex biochemical networks of redox balance and antioxidation defenses in cells suggest that combinations of antioxidant compounds, each contributing specific antioxidation mechanisms, may result in additive or synergistic effects protecting against cisplatin ototoxicity [36,37]. A cocktail of vitamins A, C, and E (ACE) combined with Mg++, the latter likely acting primarily as a cochlear vasodilator, restored hearing sensitivity and outer hair cell loss more efficiently than single compounds in noise-induced hearing loss [36,37], reducing oxidative stress and apoptosis [36]. The same antioxidant combination has shown efficacy in gentamycin-induced hearing loss [38] and cochlear implant-associated loss of residual hearing [39]. Also, there is auditory threshold recovery in age-related hearing loss after long-term oral administration of ACEMg [40].

We have tested whether a similar principle may apply to antioxidant otoprotection in cisplatin ototoxicity. Despite still-inconclusive clinical evidence, it is interesting that a combination of coenzyme Q and multivitamins [41] has been the only one to show, in a small trial, significant benefit in preventing cisplatin-induced hearing loss in a recent meta-analysis of thirteen randomized trials [25].

We have used N-acetylcysteine (NAC) and/or ACE and Mg^++^ (ACEMg) as antioxidant otoprotectants against cisplatin ototoxicity. The rationale for the use of NAC is that, like other thiosulfates [30,42], NAC is a free radical scavenger [43]. NAC also replenishes glutathione by providing cysteine residues for its synthesis [33,44,45]. Thus, it increases levels of the key intracellular redox regulator glutathione, particularly during high demand or depletion [43]. Therefore, NAC shares free radical scavenging properties with non-enzymatic antioxidant vitamins A, C, and E (see below), and it also promotes enzymatic antioxidant defenses by regenerating a key substrate such as glutathione. Pre-clinical experimental evidence of otoprotection [46,47,48], along with its wide availability and limited adverse effects, make NAC a potentially useful otoprotective antioxidant, although evidence for clinical use in cisplatin ototoxicity still is inconclusive [25].

As far as ACEMg is concerned, vitamin A, essentially in the form of carotenoids, scavenges free radicals and non-radicals in lipid environments, notably singlet oxygen and peroxyl radicals [49]. Carotenoids tested individually against cisplatin ototoxicity include lutein [50,51,52] and lycopene [53,54,55]. β-carotenes are powerful neutralizers of lipid-derived peroxyl radicals due to abundant conjugated double bonds [36,56]. They have not been tested systematically in cisplatin ototoxicity. Vitamin E, especially as α-tocopherol, traps lipid peroxide radicals, interrupting the chain reaction of lipid peroxidation, thus protecting membrane phospholipids [36,57,58,59]. Tocopherol and its esterified form tocopheryl acetate protect against cisplatin ototoxicity in rodents by limiting oxidative stress [60,61,62], and so does the water-soluble analog Trolox, 6-hydroxy-2,5,7,8-tetramethylchroman-2-carboxylic acid [63]. Vitamin C mainly detoxifies hydrogen peroxide (H_2_O_2_) in aqueous interphases through the glutathione-ascorbate cycle. It has also shown effectivity against cisplatin ototoxicity in rodents [62,64]. Therefore, vitamins A, C, and E protect against excess oxidative stress by complementing their scavenging properties in lipid (vitamins A and E) or aqueous (vitamin C) cellular environments. As mentioned above, Mg^++^ seemingly contributes to increased cochlear availability of vitamins through vasodilation and maybe also antioxidation properties [36,37]. Potentiation of combined ACEMg antioxidant otoprotection seen in noise-induced hearing loss [36,37] supports experimental testing against cisplatin ototoxicity. As explained above, in the complex cellular network of antioxidation routes, NAC may provide additional mechanisms to counteract oxidative stress. Thus, it is relevant to investigate whether NAC, ACE and Mg^++^ potentiate their antioxidant otoprotection capabilities against cisplatin ototoxicity.

Towards this goal, we applied subcutaneous injections of different doses and/or combinations of vitamins ACE, Mg^++^, and NAC to test efficacy and cellular mechanistic correlates of antioxidant otoprotection against cisplatin ototoxicity in a rat experimental model. Auditory thresholds in ABRs, OHC cell counts, and semi-quantitative immunocytochemistry for the oxidative stress marker 3-nitrotyrosine were used to compare otoprotective effects among different treatments.

## 2. Materials and Methods

### 2.1. Experimental Animals

Fifty-five male Wistar rats aged 11 to 13 weeks (Charles River Laboratories, Barcelona, Spain) were included in the experiments. Rats were kept in quarters with controlled temperature and humidity conditions, with a 12 h light-dark cycle and free access to water and chow, at UCLM Animal House in Albacete (Spain). Spanish (Royal Decree 53/2013 and Law 32/2007) and European Union (Directive 2010/63/EU) regulations on the matter of the protection and care of animals used for scientific purposes were followed. Experimental protocols were reviewed by an “ad hoc” institutional ethics committee and approved by the competent authority (official registry code, PR-2016-04-11).

### 2.2. Ototoxicity Induction by Cisplatin and Otoprotection Treatment Groups

Animals were randomly assigned to the experimental groups shown in Table 1. Ototoxicity in groups II-IX (Table 1) was induced with a single intraperitoneal injection of cisplatin (Sigma-Aldrich/Merck, Madrid, Spain, catalog number P4394) at 16 mg/kg, dissolved in 0.9% saline at 1 mg/mL. After the injection, animals were allowed to survive for 72 h, when ototoxicity is fully established [65,66]. Saline vehicle control was designated as Group I.

Otoprotective treatments in the corresponding experimental groups (Table 1) were carried out by daily subcutaneous injection of the corresponding antioxidant compound, individually or in combinations shown in Table 1, initiated 5 days prior to the induction of cisplatin ototoxicity, for a total of 7 days.

Doses of each compound were as follows: NAC (Sigma-Aldrich, Cat No. A9165) at 500 mg/kg/day (NAC500) and 1000 mg/kg/day (NAC1000); vitamin A (100 mg/kg/day as beta-carotene, Sigma-Aldrich, Cat. No. C9750); vitamin C (500 mg/kg/day, Sigma-Aldrich, Cat. No. A92902); vitamin E (200 mg/kg/day, in the form of Trolox, Sigma-Aldrich, Cat. No. 238813); MgSO_4_ (343 mg/kg/day, Sigma-Aldrich, Cat. No. M2643). Vitamin A was dissolved in sesame oil (Sigma-Aldrich, Cat. No. S3547) at a concentration of 58 mg/mL, vitamin E (Trolox) in DMSO at a concentration of 220 mg/mL, and vitamin C and MgSO_4_ in 0.9% saline at 30 mg/mL and 20.6 mg/mL, respectively. Combinations of hydro-soluble compounds were administered in the same injection, whereas vitamins A and E were administered separately in their respective solvents. Dose selection and treatment protocols were based on available literature on the protective effects of the different antioxidant compounds on cisplatin toxicity in rats, using injection as the administration route [67,68,69,70,71]. Duration of the exposure to the antioxidant compounds was empirically based on published results with injections of ACEMg as otoprotectant against noise-induced hearing loss [37].

Animals were distributed in nine groups (see Table 1). Group I were control rats injected intraperitoneally with a volume of saline equivalent to the volume used for cisplatin administration, in addition to a subcutaneous administration of a volume of saline, sesame oil, and DMSO equivalent to that used for the administration of the corresponding otoprotective treatments. Group II were rats injected intraperitoneally with cisplatin. Groups III to IX were treatment groups with different doses and/or combinations of otoprotective agents, as detailed in Table 1. Animals were sacrificed the day after the last administration, following auditory brainstem response (ABRs) recordings.

### 2.3. Auditory Brainstem Response Recordings (ABRs)

ABRs were recorded as described previously in detail [72,73]. Briefly, rats were anesthetized with isoflurane (induction at 4%, maintenance at 1.5–2%, 1 L/min O_2_ flow rate) and placed in a sound-insulated chamber (Incotron Eymasa S.L., Barcelona, Spain). Body temperature was maintained (37.5 °C) with a thermal pad and monitored with a rectal probe. Recording electrodes (Rochester Electro-Medical, Tampa, FL, USA) were inserted below the skin at the right mastoid apophysis for the inverted, at the cranial vertex or the non-inverted, and at the left mastoid apophysis for the ground. BioSig System III from Tucker-Davis Technologies (TDT, Alachua, FL, USA) was used for stimulation and recording. Tone burst stimuli (5 ms rise/fall no plateau, 20 repetitions/second at 0.5, 1, 2, 4, 8, 16, and 32 kHz) were built with SigGenRP software v. 4.4 (TDT) and an RX6 Piranha Multifunction Processor hardware (TDT). Tones were delivered into the right ear external canal through an EC-1 electrostatic speaker with an ED1 controller (TDT). SigCalRP software v. 4.2 (TDT) and an ER-10B+ low-noise microphone (Etymotic Research Inc., Elk Grove, IL, USA) were used for stimuli calibration.

To obtain auditory thresholds, responses were recorded in 5 dB decremental steps starting at 80 dB SPL, which was set as a safe upper limit for noise overstimulation confounders [72,74]. The stimulus intensity producing wave responses with peak-to-peak amplitudes larger than 2 standard deviations (SD) of the background activity, measured prior to the tone burst onset, was taken as the auditory threshold [72,73].

The change in auditory threshold relative to the control group for each of the frequencies analyzed was determined by subtracting post-treatment auditory thresholds from auditory thresholds before treatment instauration. Thresholds in Group I were used as the reference baseline. Group VI- NAC500 + ACE + Mg was not considered for the statistical analysis of threshold changes, since stable records of ABRs were obtained only from a single subject.

### 2.4. Cochlear Fixation and Processing for Histology

Rats received an intraperitoneal overdose of sodium pentobarbital (Dolethal, Vetoquinol, Madrid, Spain; i.p. 700 mg/kg) for euthanasia. Intravascular perfusion fixation was started by flushing with 0.9% saline, 5 min, followed by 4% p-formaldehyde (PFA 4%) in 0.1 M phosphate buffer (PB, pH 7.3), 15 min. Cochleae were then dissected, rinsed in phosphate-buffered saline (PBS), 3 × 5 min, and decalcified in 50% RDO (rapid decalcifying solution, Apex Engineering Products Corporation, IL, USA). The right cochleae, for quantification of outer hair cells (OHCs) by fluorescent immunolocalization of myosin VIIA as detailed below, were kept in RDO for 1 h, while the left ones, used for 3-nitrotyrosine (3-NT) immunolocalization (a marker of oxidative stress, see below), were kept in RDO for 2 h. After incubation in RDO, the cochleae were washed again 3 × 5 minin PBS. The concentration of RDO and the exposure time were adjusted in pilot experimental runs so that decalcification did not interfere with fluorescence labeling.

### 2.5. Immunohistochemistry on “In Toto” Cochlear Surface Preparations for Outer Hair Cell Quantification

Decalcified cochleae immersed in PBS were dissected using an Olympus SZX10 stereoscopic microscope (Olympus, Tokyo, Japan). With two thin-tipped tweezers and a surgical blade, a cut perpendicular to the long axis of the modiolus was made, dividing the cochlea in two segments of equal length, one more apical and one more basal. The otic capsule was excised, and lateral wall structures, the modiolus and tectorial membrane, were dissected out and removed, thus exposing the organ of Corti. Cochlear turn fragments (between 2 and 5 per full cochlear length) were placed in PBS in separated wells of a multi-well plate (Sigma-Aldrich, catalog # CLS3736). To identify OHCs, fluorescent immunolocalization of myosin VIIa was performed. Segments were placed in PBS plus 1% Triton X-100 and 5% bovine serum albumin (PBS-1% Tx100-5% BSA) for 1 h at RT, followed by overnight incubation (4 °C) with anti-myosin-VIIa primary antibody (Proteus BioSciences, catalog no. 25-6790) in PBS-0.2% Tx100-3% BSA at a concentration of 1:1000. The primary antibody was then discarded and three 5 min washes with PBS were carried out, followed by incubation in anti-rabbit Alexa Fluor 594 secondary antibody (Thermo Fisher Scientific, catalog no. A-21207) in PBS-0.2% Tx100-3% BSA at a concentration of 1:200 for 1 h at RT in the dark. Following 3 × 5 min washes with PBS, immunostained cochlear turn fragments were mounted on slides with Vectashield containing DAPI (Vector Laboratories, Burlingame, CA, USA). Such “surface preparations” were analyzed and photographed on a Zeiss 710 confocal microscope (Zeiss, Oberkochen, Germany). After image capture, ImageJ software (National Institutes of Health, USA) was used to visualize OHCs and count loss for each 0.1 mm of length of the organ of Corti as described below.

### 2.6. Immunohistochemistry for 3-Nitrotyrosine in Cochlear Sections

After decalcification and washing, the left cochleae were submerged in 30% sucrose (Panreac, Barcelona, Spain, catalog number 131621) in PB for 48–72 h for cryoprotection. Subsequently, they were embedded in gelatin blocks (Panreac, catalog number 142060) prepared in PB-30% sucrose. Blocks were deep-frozen at −70 °C using 2-propanol (Sigma-Aldrich, catalog number I9516) on dry ice and stored at −80 °C until further use. From each block, 20 µm thick paramodiolar sections were made using a Leica CM3050 S cryostat (Leica, Wetzlar, Germany). Sections were collected on Superfrost^R^ slides and stored at −80 °C. For detection of 3-NT, cochlear sections were tempered at RT for 30 min and post-fixed with 4% PFA for 8 min. After 3 × 5 min washes with PBS, 1 mL PBS-0.25% Tx100-2% BSA was added to the slides to block non-specific binding and for permeabilization, for 1 h at RT. Slides were then exposed overnight to a mouse anti-3-nitrotyrosine monoclonal antibody (Abcam ab61392) at a 1:100 dilution at 4 °C. The next day, after 3 × 5 min washes with PBS, sections were incubated with a donkey anti-mouse Alexa 488 secondary antibody (1:200, Thermo Fisher Scientific A-21202) for 1 h at RT in the dark. Antibody dilutions were made in PBS-BSA 0.5%. After rinsing 3 × 5 min with PBS, slides were cover slipped with Vectashield with DAPI and analyzed with laser scanning confocal microscopy in a Zeiss 880 microscope. Immunofluorescence intensity was quantified using ImageJ (National Institutes of Health), as detailed below.

### 2.7. OHC Counts

For OHC counts, ImageJ software (National Institutes of Health, MD, USA) was used. Segments of each cochlea, digitally captured with the confocal microscope, were ordered from apical to basal. Segments of 0.1 mm were measured along a line delimiting the apical portion of the inner hair cells using the ImageJ “segmented line” tool. In each segment, the percentage of identifiable OHCs was determined as previously described [66]. This allowed its representation in the form of “cytocochleograms” and the analysis of the damage in the organ of Corti, quantifying the length of 3 well-differentiated zones along the cochlear turn: a zone with 100% of OHCs, a transition zone (with a partial loss of OHCs), and a zone with 0% of OHCs [66].

### 2.8. Semiquantitative Measurement of Fluorescent Signal Intensity for 3-Nitrotyrosine

Semiquantitative analysis of immunofluorescence for 3-nitrotyrosine (3-NT) was also carried out using ImageJ. Confocal images captured from paramodiolar cochlear sections (see Section 2.6 above) were used. To measure the intensity of the fluorescence signal for 3-NT, the ImageJ “polygon selections” tool was used. With it, contours of OHCs visible in the apical turn were selected, excluding the region occupied by the nucleus. Next, the mean value of the signal intensity inside the selected contour was calculated, as previously detailed [66].

### 2.9. Statistical Analysis

The Kolmogorov–Smirnov test was applied to determine normal distribution of data. After that, a one-way analysis of variance (ANOVA) and a post hoc test (Tukey) was carried out to compare the means of the data from the different study groups for the three metrics utilized, i.e., threshold shifts in ABRs, OHC counts, and relative intensity of 3-NT immunolabeling. Statistical analysis was carried out with SPSS 25.0 software (IBM, Armonk, NY, USA). Statistical significance was graphically expressed as * *p* < 0.05, ** *p* < 0.005, *** *p* < 0.001.

## 3. Results

### 3.1. Auditory Threshold Shifts after Different Antioxidant Treatments

In contrast with non-recordable activity and thus undetectable thresholds in Group II-Cis rats, activity was recordable with all antioxidant treatments to a lesser or greater extent and therefore thresholds and threshold shifts were within detectable ranges throughout all tested frequencies. However, different antioxidant treatments differed in threshold shifts across frequencies, pointing to different recovery trends towards control and baseline thresholds (Figure 1, Table 2). For this reason, threshold shifts were analyzed relative to Group I-Control.

The smallest threshold shift values relative to Group I-Control were found in group IX-ACEMg, which showed no statistically significant differences with Group I-Control at any tested frequency (Figure 1). In Group VIII-Mg, threshold shifts were also statistically not significantly different from Group I-Control at frequencies of 8 kHz and below. However, unlike Group IX-ACEMg, significantly elevated threshold shifts were found relative to Group I-Control at 16 and 32 kHz (*p* < 0.005, *p* < 0.05, respectively) (Figure 1). In Group III-NAC500, shifts in threshold were not statistically significant at 4 kHz and below, whereas shifts at 8 kHz (*p* < 0.05), 16 kHz (*p* < 0.005), and 32 kHz (*p* < 0.05) were so (Figure 1, Table 2). In Group VII-ACE, threshold shifts were not significant at 2 kHz and below, whereas, different from Group III-NAC500, significant shifts in threshold were recorded at frequencies of 4 kHz and above (*p* < 0.05, *p* < 0.001, *p* < 0.001, *p* < 0.001, respectively) (Figure 1). Group IV-NAC1000, Group V-NAC500 + Mg, and Group VI-NAC500 + ACEMg showed the largest average threshold shifts across frequencies and thus a lower trend towards threshold recovery to control values, especially from 2 kHz upwards (Figure 1, Table 2). In the case of Group IV-NAC1000, significant threshold shifts were observed at 2, 4, 8, 16, and 32 kHz (*p* < 0.05, *p* < 0.05, *p* < 0.001, *p* < 0.001, *p* < 0.001, respectively), and in Group V-NAC500 + Mg, virtually across the whole frequency range, specifically at 0.5, 1, 2, 8, 16, and 32 kHz (*p* < 0.05, *p* < 0.05, *p* < 0.05, *p* < 0.05, *p* < 0.005, *p* < 0.005, respectively) (Figure 1).

On the other hand, there were statistically significant differences between Group IX-ACEMg and Group VII-ACE at 1 kHz (*p* < 0.05) and 16 kHz (*p* < 0.05). Likewise, statistically significant differences were observed between Group IX-ACEMg and Group VIII-Mg, at least at 1 kHz (*p* < 0.05) and at 16 kHz (*p* < 0.05). Finally, in Group VI-NAC500 + ACEMg, ABR recordings with identifiable waves were obtained only from a single subject (Figure 1, Table 2). The remaining animals did not show recordable activity, which precluded statistical analysis of threshold shifts.

### 3.2. Outer Hair Cell Counts

Cytocochleograms (Figure 2) showed a pattern of OHC loss characterized by three regions or segments [65]. Animals in Group II-Cis presented an apical portion, representing an average of 32.8% ± 3.28 of cochlear length, with complete preservation of OHCs (zone with 100% of OHCs). In the basal portion representing an average of 32.7% ± 11.31 of cochlear length, there was complete loss of OHCs (zone with 0% of OHCs), whereas in the third portion, located between the previous ones, representing 34.4% ± 11.77 average of cochlear length, there was partial loss of OHCs (Figure 3).

In treatment Groups VII-ACE, VIII-Mg, and IX-ACEMg, the most apical segments of the organ of Corti, with 100% OHC preservation, were significantly longer than in Group II-Cis, representing, respectively, 65.75% ± 24.75 (*p* < 0.05), 72.47% ± 31.93 (*p* < 0.05), and 55.45% ± 13.47 (*p* < 0.05) on average (Figure 2 and Figure 3). In contrast, in treatment Group III-NAC500, Group IV-NAC1000, Group V-NAC500 + Mg, and Group VI-NAC500 + ACEMg, the average percentages of the most apical length of the organ of Corti with 100% OHCs were, respectively, 50.32% ± 30.46, 35.43% ± 6.17, 48.32% ± 29.13, and 36.41% ± 10.05. These differences were not statistically significant from values in Group II-Cis (Figure 3).

A significantly shorter basal length of the organ of Corti with 0% OHCs compared to Group II-Cis was found in Group III-NAC500, Group V-NAC500 + Mg, Group VII-ACE, Group VIII-Mg, and Group IX-ACEMg, with average values of 14.71% ±9.53 (*p* < 0.05), 11.37% ± 8.72 (*p* < 0.05), 3.61% ± 5.06 (*p* < 0.001), 6.45% ± 9.27 (*p* < 0.005), and 10.70% ± 4.66 *p* < 0.05), respectively (Figure 2 and Figure 3). Group IV-NAC1000 and Group VI-NAC500 + ACEMg showed a basal length of the organ of Corti with 0% OHCs of 25.26% ± 7.54 and 18.65% ± 8.54, respectively, statistically not significantly different from Group II-Cis (Figure 2 and Figure 3).

Finally, the average percentage of the organ of Corti occupied by the intermediate zone (“transition zone” [65], with partial preservation OHCs, was 34.96% ± 22.65 in Group III-NAC500, 39.31% ± 11.03 in Group IV-NAC1000, 40.30% ± 22.79 in Group V-NAC500 + Mg, 44.95% ± 15.94 in Group VI-NAC500 + ACEMg, 30.64% ± 22.43 in Group VII-ACE, 21.07% ± 24.86 in Group VIII-NACMg, and 33.85% ± 12.34 in Group IX-ACEMg. None of these length differences showed statistical significance relative to Group II-Cis (Figure 2 and Figure 3).

### 3.3. 3-Nitrotyrosine Immunolabeling for Oxidative Stress

As far as 3-NT immunolabeling is concerned (Figure 4A), in regions with identifiable OHCs, Group II-Cis samples showed a significant increase in 3-NT immunoreactivity intensity of 87.90% (*p* < 0.05) relative to Group I-Control (Figure 4B). Treatment Group IV-NAC1000 also showed a significant increase in immunolabeling intensity of 160.00% (*p* < 0.001) relative to Group I-Control (Figure 4A,B). In contrast, in treatment Group V-NAC500 + Mg, Group VII-ACE, Group VIII-Mg, and Group IX-ACEMg, there were significant reductions in 3-NT immunoreactivity of 51.76% (*p* < 0.05), 54.92% (*p* < 0.05), 44.81% (*p* < 0.05), and 56.41% (*p* < 0.05), respectively, relative to Group II (Figure 4A,B). These values did not differ statistically from those of Group I-Control (Figure 4B). Finally, in Group III-NAC500, relative 3-NT immunolabeling intensity was closer to Group I-Control, whereas in Group VI-NAC500 + ACEMg it was closer to Group II-Cis, although neither of these values were statistically significant (Figure 4A,B).

## 4. Discussion

We report here that limiting the cochlear oxidative stress response of cisplatin with different antioxidant combinations attenuates the loss of hearing sensitivity and limits toxic structural damage to the auditory receptor organ. Oxidative stress, tested with immunocytochemical detection of the oxidative stress marker 3-NT, OHC survival rate, assessed with OHC counts, and level of hearing threshold recovery, assessed with ABR recordings, all vary with each antioxidant or antioxidant combination utilized. Considering the overall outcome of experimental tests among different formulations, the ACEMg combination, 500 mg NAC, and MgSO_4_ alone seem to preserve better auditory thresholds and/or OHC survival after cisplatin ototoxicity. Of all these, however, ACEMg was the only one simultaneously showing threshold shifts not significantly different from normal baseline thresholds at any tested frequency, large preservation of OHCs, and oxidative stress levels closer to normal values, as seen with 3-NT immunolabeling. In contrast, high doses of antioxidants, such as in the NAC1000 experimental group, or extensive combinations, such as NAC500 with vitamins A, C, and E and MgSO_4_, resulted in lower attenuation of cisplatin-induced oxidative stress, poorer survival of OHCs, reduced recovery from threshold shifts, or a combination thereof. This suggests that once an “antioxidation threshold” is surpassed, treatments may greatly lose efficacy by contributing excess redox imbalance [75].

### 4.1. Hearing Loss and Cochlear Damage after Cisplatin Ototoxicity in the Rat Model

Under our experimental conditions, rats in the cisplatin-exposed group showed complete loss of evoked activity in ABRs, with undetectable hearing thresholds at all tested frequencies up to the highest applied intensity of 80 dB SPL. Increased thresholds throughout the whole frequency range, at the limit of precise detection, have been reported previously using similar cisplatin exposure protocols [65,66]. Basal OHCs are more prone to cisplatin ototoxicity and usually die faster after exposure [14], which correlates with intrinsically lower levels of the antioxidant glutathione [76]. Therefore, total or partial loss of OHCs at the cochlear base after cisplatin exposure, comprising around 67% length in our experimental series, correlates with highly elevated hearing thresholds in the high-frequency range [77,78]. In contrast, more apical OHCs localized in a segment of about 33% of cochlear length in the cisplatin-exposed group are more resistant to toxic damage [5,14,66,79]. Therefore, threshold elevations detected in lower frequency ranges after cisplatin ototoxicity may be due in part to compromised electrical signal generation by otherwise surviving apical OHCs [65,66]. The stria vascularis generates the endocochlear potential, the driving force for electrical activity in OHCs. It is particularly sensitive to cisplatin accumulation and thus oxidative stress damage [17]. Immunolabeling for the oxidative stress marker 3-NT is more widespread in the stria vascularis after cisplatin exposure than with other ototoxic agents such as kanamycin [66]. A single dose of 16 mg/kg in Wistar rats, the same used in the present work, reduces the endocochlear potential by around 50% [80].

In sum, regardless of the dose-dependency of cisplatin ototoxicity, with progressive damage starting at the most sensitive basal/high-frequency cochlear end [77,78], experimental doses like the one used in this work may lead to simultaneous, significant threshold elevations across the whole frequency range. This is likely due to a combination of progressive OHC loss, starting at the cochlear base, and early widespread damage to the stria vascularis, including the cochlear apex [17,66]. Therefore, this experimental model in rat reproduces acute, extensive cisplatin-mediated ototoxicity. Such a model of extreme ototoxic damage provides an opportunity to explore the effectiveness of antioxidant otoprotection mechanisms.

### 4.2. Antioxidants and Antioxidant Combinations in Otoprotection against Cisplatin Ototoxicity

As already mentioned, cochlear oxidative stress, originating from overproduction of ROS/RNS and other highly reactive free radicals and non-radicals, is one of the main pathophysiologic mechanism of hearing loss induced by multiple toxic insults, including noise and ototoxic drugs [81,82,83], notably aminoglycoside antibiotics or antineoplastic chemotherapeutic agents such as cisplatin [66,84,85].

We have used three metrics, namely threshold shifts in ABRs, OHC counts, and immunocytochemical labeling with the oxidative stress marker 3-NT, to provide comparative evidence of the experimental efficacy of different antioxidants and antioxidant combinations against cisplatin ototoxicity.

As previously mentioned in relation to ABRs, there was no detectable activity at the tested frequencies in the group treated with cisplatin, thus precluding threshold shift calculations. In contrast, to a greater or lesser extent, activity was recordable in all experimental antioxidant treatment groups. Because of this, results gain context by comparing threshold shifts after treatments relative to controls. This shows trends of different treatments towards recovery of normal “physiological” baseline thresholds. In this regard, ACEMg treatment resulted in threshold shifts which were closer to normal baseline auditory threshold levels than any other antioxidant compound tested. Threshold shifts after ACEMg injections did not differ statistically from control baseline values across the entire tested frequency range. In contrast, threshold shifts recorded after administration of ACE without MgSO_4_ differed statistically from baseline values at frequencies above 2 kHz and were statistically significantly higher than those of ACEMg, at least in part of the frequency range. This supports the idea that Mg^++^ potentiates ACE antioxidant otoprotection. In fact, ACEMg has been previously shown to provide superior preservation of hearing sensitivity than ACE or MgSO_4_ alone in noise-induced hearing loss in guinea pigs [37], likely through cochlear vasodilation [36,86]. ACEMg also protects against gentamycin-induced hearing loss in guinea pigs [38] and age-related hearing loss in rats [40]. Here, we provide evidence of strong otoprotection with ACEMg against cisplatin ototoxicity. However, different to noise-induced hearing loss [37], extreme loss of hearing sensitivity after cisplatin ototoxicity also was significantly attenuated by injections of MgSO4 alone, although, in contrast with ACEMg, shifts at higher frequencies still had statistically significantly higher values than baseline. Also, threshold shifts were significantly higher than those of ACEMg, at least in part of the tested frequency range.

It is unclear how Mg^++^ by itself may improve hearing loss in cisplatin ototoxicity [87]. There is hypomagnesemia after cisplatin treatment, and although a Mg^++^-enriched diet failed to provide histological otoprotection in guinea pigs [88], evidence of recovery of otoacoustic emissions after cisplatin treatment supplemented with Mg^++^ has been reported [87,89]. Vasodilator, antioxidant, and ion homeostatic effects of Mg^++^ may critically improve the metabolic status of the stria vascularis, a main target of cisplatin ototoxicity, one possibility which requires further experimental testing. However, therapeutic handling of Mg^++^ in ototoxicity may be complicated in humans by relatively frequent occurrence of diarrhea [39]. No diarrhea was evidenced in our experimental animal series, which may assist in working out dosing for potential applications.

After treatment with 500 mg NAC (NAC500 group) daily for seven days, with cisplatin ototoxicity induced at day 5, threshold shifts at frequencies of 4 kHz and below had average values not significantly different from baseline. However, threshold shifts remained significantly elevated at 8 kHz and above. We then sought to test whether, in combination with MgSO4 (NAC500+Mg group), there were additionally diminished threshold shifts across frequencies, suggesting additive/synergistic otoprotection between both compounds. In the NAC500 + Mg group, threshold shifts remained significantly elevated above baseline across the entire frequency range. Therefore, as far as auditory thresholds are concerned, Mg^++^ does not seem to potentiate NAC otoprotective antioxidation in the cochlea after cisplatin ototoxicity. Rather, it seems to decrease it. A rather similar result was obtained with a higher NAC dose of 1000 mg/Kg (NAC1000), also administered daily for 7 days. Threshold shifts remained significantly elevated at 2 kHz and above. We speculate that high doses of NAC or increased cochlear availability of the compound when combined with MgSO_4_, due to Mg^++^-induced cochlear vasodilation, probably along with additional Mg^++^-induced antioxidation, may generate excess reduction potential within cells, with redox imbalance leading to “anti-oxidative stress” and cell damage triggered by disruption of physiological ROS/RNS signaling [75]. In other words, our findings suggest a therapeutic “antioxidation threshold”, above which antioxidant otoprotection probably provides no benefit. This was also seen when combining antioxidation with NAC and ACEMg (see introduction, [90,91]). Treatment with a cocktail of NAC500 and ACEMg resulted in undetectable thresholds in five out of six rats exposed to cisplatin. Only in one rat of this group were auditory thresholds recorded. In this individual animal, representing a “best case” outlier, threshold shifts did not differ much from those in the NAC1000 group, particularly at frequencies of 8 kHz and above. Because no activity was recordable in the remaining animals of this treatment group, average threshold shifts were beyond detection and analysis. Therefore, NAC500 combined with ACEMg seems to have overall comparatively poorer otoprotectant capacity. This is an additional indication that antioxidant excess may critically hamper hearing sensitivity preservation from cisplatin ototoxicity.

Survival of OHCs offers additional insights into antioxidation protection against cisplatin ototoxicity. In general, OHC survival rate, from counts on surface preparations of the organ of Corti, corresponds reasonably well with threshold shift recoveries after the different treatments. Tested treatments showed improved OHC survival relative to the extensive OHC loss seen in cisplatin-treated animals, except for the NAC1000 dose and the NAC500 + ACEMg cocktail. ACEMg, MgSO4, and ACE treatments resulted in increased OHC survival. Improved OHC survival in the ACE-treated group is somehow in contrast with still significantly elevated threshold shifts at frequencies of 4 kHz and above in this treatment group. This suggests that OHC survival may not be sufficient indicator of hearing sensitivity preservation after antioxidant protection against cisplatin ototoxicity. For instance, compared with other compounds, ACE might not promote sufficient recovery of the stria vascularis and related structures of the lateral wall, with the corresponding consequences for OHC function. It has been shown that in mice with noise-induced hearing loss, an oral formulation highly enriched in vitamin A, C, and E and Mg^++^ preserves the cellular organization of the cochlear lateral wall over a less-enriched formulation [92], supporting the idea that protection of the stria vascularis with antioxidants depends on the composition and concentration of the formulation. OHC survival was also significantly increased with both NAC500 and NAC500 + MgSO4 treatments, although, different to the treatments previously discussed, the apical cochlear segment with maximum survival of OHCs [66] was not significantly lengthened, and increased survival of OHCs was mostly at the expense of the reduction in the length of damage in the basal segment, where OHCs undergo massive loss after cisplatin exposure [66]. This may indicate slower or reduced OHC recovery rates in the latter two treatments.

There were no statistically significant differences in OHC loss among cisplatin-exposed rats and rats treated with NAC1000 or NAC500 + ACEMg. Lack of significant OHC survival enhancement matches limited or no threshold shift reductions with these same two treatments, specially NAC500 + ACEMg, after which the vast majority of cisplatin-exposed rats continued to lack evoked activity. Residual OHC survival, even below statistical significance, along with lower or higher level of damage to the stria vascularis may determine whether activity is limited or not recordable. Again, high doses or large antioxidant compound combinations might be pushing redox homeostasis out of balance [75], limiting cell survival recovery. This hypothesis warrants further testing, more so considering that several antioxidants and pro-oxidants induce hair cell loss in “in vitro” assays in cochlear micro-explants [93]. It is also relevant that high concentrations of trans-tympanic NAC caused greater alterations in the inner ear than those produced by administration of cisplatin itself [94].

Immunocytochemical detection of the oxidative stress marker 3-NT provided further insights into the antioxidant otoprotection properties of the different treatments tested. Because we were interested in spatial distribution of immunolabeling, we measured relative levels of 3-NT immunostaining in selected regions with preserved OHCs. 3-NT labels protein nitrosylation at tyrosine residues, a specific indicator of oxidative stress mediated by RNS, specifically peroxynitrites [66,95]. In the cochleae of cisplatin-exposed rats, 3-NT immunolabeling was significantly more intense relative to vehicle-injected controls. In regions with preserved, identifiable OHCs, relative immunolabeling intensity was almost twice than in controls, indicating increased oxidative stress even in surviving OHCs after cisplatin treatment. Such elevated 3-NT levels were significantly reversed with all tested treatments, except for NAC1000 and NAC500 + ACEMg. In the latter group, 3-NT intensity levels were indistinguishable from those in the cisplatin exposed group. In the former, relative 3-NT levels showed a trend to be even higher than in the cisplatin-exposed rats, although without reaching statistical significance. When reversed by treatments, 3-NT immunolabeling returned to relative intensity values significantly lower than those in cisplatin-exposed rats. Except for the NAC500 group, which did not reach statistical significance, the rest of treatments lowered cochlear oxidative stress increased by cisplatin, somehow mirroring OHC preservation. In contrast, NAC1000 and NAC500 + ACEMg treatments, which provide limited, statistically not significant OHC preservation and limited threshold shift reductions, do not significantly reverse cisplatin-induced oxidative stress, at least in surviving OHC regions. This adds to the notion of excess therapeutic “antioxidant power” likely resulting in failure of antioxidation mechanisms. In this regard, however, it is interesting to remark that NAC500 + MgSO4 resulted, after cisplatin exposure, in still significantly elevated threshold shifts virtually throughout the entire tested frequency range, but, at the same time, with significant preservation of OHCs, along with low levels of oxidative stress. This points to additional mechanisms of cochlear disfunction induced by excess antioxidants which require further research. One possibility is that the NAC500 + Mg formulation may promote antioxidation and limited OHC survival comparable to that of NAC500 (see Figure 3 and Figure 4) while negatively affecting the stria vascularis. This possibility of differential sensitivities to oxidative stress awaits further study.

### 4.3. Limitations of the Study

Due to evolutionary proximity, essential mechanisms of toxic damage affecting auditory function in humans, as well as otoprotection, have been unraveled in rats and other rodents at the cellular and molecular levels [96,97]. However, besides similarities, there are also differences, mostly at the organ and whole organism levels [98], which may limit direct translation to the human clinical setting. Differences in cochlear anatomy and physiology [98] or in metabolism and pharmacokinetics [98], to name a few, may result in differences in ototoxic response to cisplatin, particularly in the timeline, extent, and pattern of ototoxic damage. For instance, rodents including rats usually require comparatively high doses of cisplatin to affect hearing [99]. This is not necessarily a limitation when searching for basic cellular mechanisms. Animal models reproducing antioxidant otoprotection against “extreme” cisplatin ototoxicity, like the one used in this and previously published studies [66], provide important mechanistic insights, crucial to accelerating the generation of translational models closer to the human setting [100].

Also, experimental injections allow researchers to fulfill the key objective of reasonable, variable-controlled proof-of-principle. However, there are still potential confounding sources. For instance, the cocktail of vitamins ACE contains DMSO as a solvent for vitamin E, which itself has free radical capture properties [101], so it cannot be completely ruled out that additional antioxidation may have had an effect on the results. Even more importantly, as far as the human translational setting is concerned, systemic administration of otoprotective antioxidants may limit the antineoplastic effects through unwanted “antioxidant protection” of tumoral cells. This emphasizes that the principles supported in this work will have to be aligned with the results of further experiments aimed at determining optimal administration routes for antioxidant otoprotection so that interference with cisplatin oncotherapy itself is avoided or limited [102].

### 4.4. Conclusions

Formulations of compounds which combine different antioxidant capacities, along with a likely relevant vasodilator effect of Mg++, such as ACEMg, show outstanding otoprotection against extreme experimental cisplatin ototoxicity by limiting oxidative stress damage. Individual compounds such as NAC show relatively more limited, although still significant, otoprotection. However, high doses or large antioxidant combinations may critically override a “threshold” redox balance, beyond which antioxidant otoprotection fails and ototoxic damage remains or may be even potentiated.

## Figures and Tables

**Figure 1 antioxidants-13-00761-f001:**
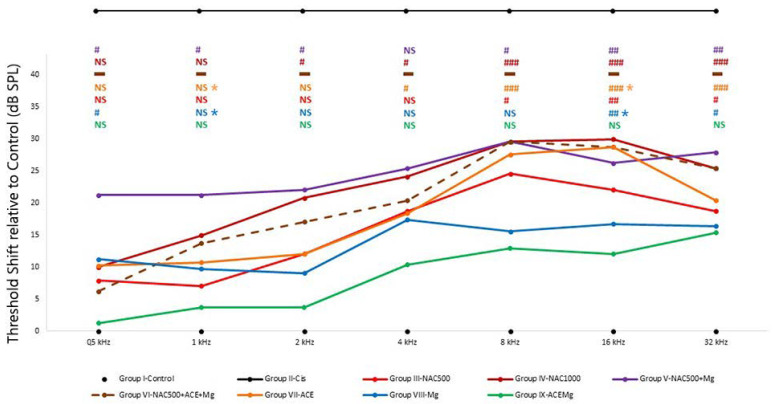
Average threshold shifts in the different antioxidant treatment groups. ABR recordings were performed 7 days after starting the corresponding treatments and 2 days after cisplatin injection. Group II-Cis has been illustrated outside the scale of the graph, representing the undetectability of auditory evoked potentials in this group at any of the frequencies or intensities studied. Overall, Group IX-ACEMg showed the smallest threshold shifts of all tested treatments. At the other end were Group VI-NAC500 + ACEMg and Group IV-NAC1000, with significant threshold shifts spanning throughout most or all tested frequencies. This suggests that excess antioxidant concentrations and/or bioavailability may override redox balance, leading to diminished antioxidant treatment efficacy. N.S.: statistically not significant *p*-values relative to normal control baseline in Group I-Control. Significant *p*-values relative to normal control baseline in Group I-Control are shown as: # *p* < 0.05, ## *p* < 0.005, ### *p* < 0.001. Significant *p*-values relative to Group IX-ACEMg in Group VII-ACE are shown as blue and yellow asterisks (*), respectively (*p* < 0.05). The broken line in Group VI indicates data obtained from a single animal and therefore not subject to statistical analysis (see text).

**Figure 2 antioxidants-13-00761-f002:**
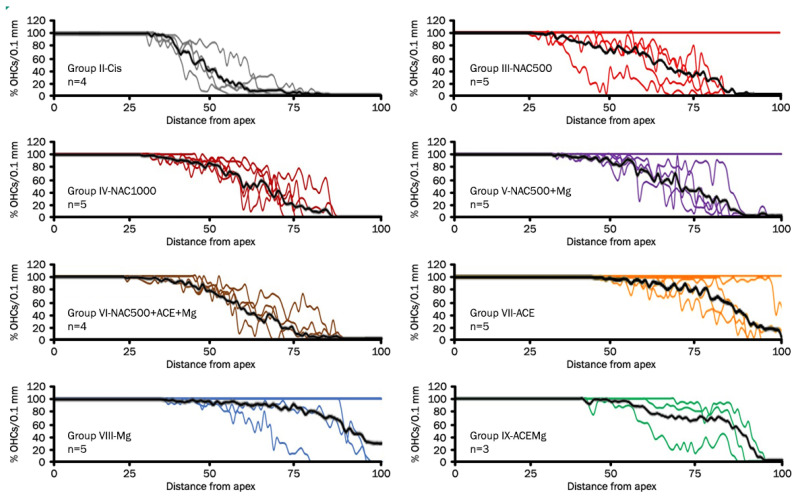
Line graphs (cytocochleograms) showing outer hair cell loss and preservation in rat cochleae from the different experimental groups. Each color line represents one cochlea, and “n” is the total number of cochleae from individual animals used for cell counts in each treatment group after eliminating defective cochlear turn samples. The black line is the average percentage of outer hair cells as a function of distance from the apex. Notice individual cases in which there is virtually no OHC loss. It is interesting that they are mostly in treatment groups providing better antioxidant otoprotection. They may represent cases of exceptional sensitivity to antioxidant otoprotection in the context of natural biological variability or, alternatively, limited sensitivity to cisplatin ototoxicity. It is worth noting that the Cis group, NAC1000, and NAC500 + ACE + Mg do not show such individual outliers.

**Figure 3 antioxidants-13-00761-f003:**
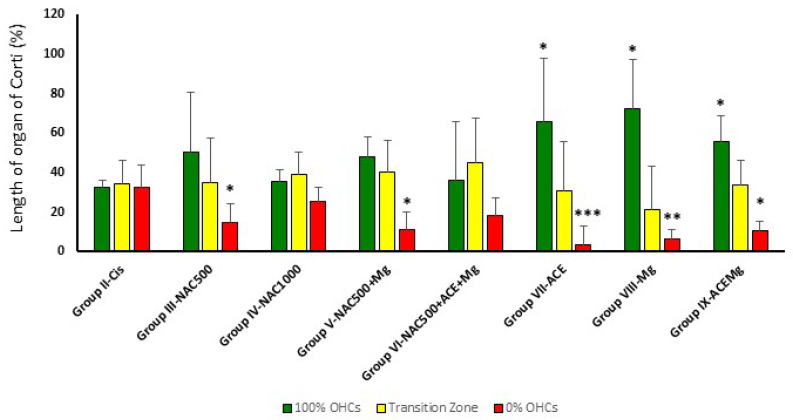
Bar graph showing the average percentage of the apical to basal length of the organ of Corti with complete preservation of OHCs (green bars), partial loss (yellow bar), or complete loss (red bar). In Group IX-ACEMg, Group VIII-Mg, and Group VII-ACE, the relative length of the organ of Corti with 100% OHC loss is significantly reduced, whereas the apical segment with maximum preservation of OHCs is longer. Notice that Group IV-NAC1000 and Group VI-NAC500 + ACE + Mg did not show significant differences with Group II-Cis in OHC survival patterns. (*) Statistical significance of *p* values relative to cisplatin, * *p* < 0.05, ** *p* < 0.005, *** *p* < 0.001.

**Figure 4 antioxidants-13-00761-f004:**
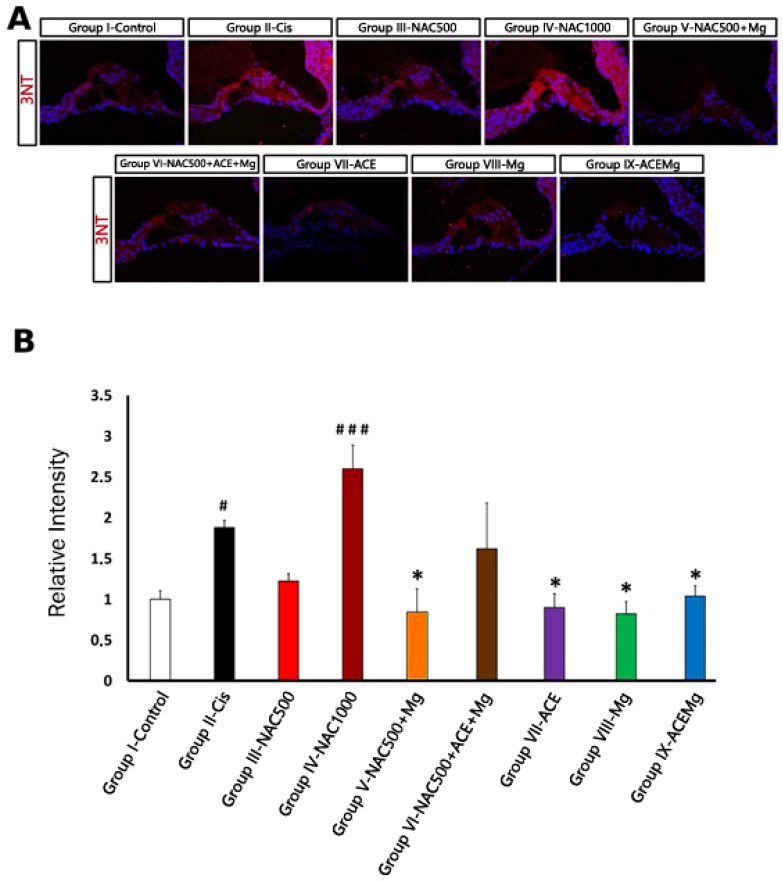
Fluorescent immunolocalization of the oxidative stress marker 3-NT in the cochlea after cisplatin ototoxicity in comparison with the different antioxidant treatments in this study. (**A**) Immunolabeling for 3-NT (red) with DAPI counterstaining (blue) in representative sections of the apical turn of the organ of Corti from rats of the different experimental groups. Notice very low or low levels of 3-NT immunolabeling in Group IX-ACEMg, Group VIII-Mg, Group VII-ACE, Group III-NAC500, and also Group V-NAC500 + Mg, comparable to Group I-control. Group IX-NAC1000 and Group VI-NAC500 + ACE + Mg show 3-NT immunostaining visually similar to Group II-Cis. (**B**) Relative intensity levels of 3-NT immunolabeling in OHC regions from cochlear sections of animals from the different experimental groups. (*) Statistical significance of *p*-values, relative to the Group II-Cis, * *p* < 0.05. (#) Statistical significance of *p*-values relative to the Group I-Control. # *p* < 0.05, ### *p* < 0.001.

**Table 1 antioxidants-13-00761-t001:** Distribution and number of animals in the different experimental groups.

Experimental Groups	Treatments	Number of Animals
I	Vehicle Control (Group I-Control)	5
II	Cisplatin (Group II-Cis)	8
III	Cis + 500 mg NAC (Group III-NAC500)	6
IV	Cis + 1000 mg NAC (Group IV-NAC1000)	6
V	Cis + NAC500 + MgSO_4_ (Group V-NAC500 + Mg)	6
VI	Cis + NAC500 + MgSO_4_ + Vitamin A, C, E (Group VI-NAC500 + ACE + Mg)	6
VII	Cis + ACE (Group VII-ACE)	6
VIII	Cis + MgSO_4_ (Group VIII-Mg)	6
IX	Cis + ACE+ MgSO_4_ (Group IX-ACEMg)	6

**Table 2 antioxidants-13-00761-t002:** Average auditory threshold shift values (dB) in the different treatment groups.

	I	III	IV	V	VI ^1^	VII	VIII	IX
**0.5 kHz**	0 ± 6.78	7.92 ± 5.77	10.00 ± 4.79	21.25 ± 0.00	6.25 ± 0.00	10.25 ± 6.52	11.25 ± 6.12	1.25 ± 5.00
**1 kHz**	0 ± 8.29	7.08 ± 2.89	15.00 ± 9.46	21.25 ± 3.54	13.75 ± 0.00	10.75 ± 2.74	9.75 ± 9.62	3.75 ± 5.00
**2 kHz**	0 ± 11.37	12.08 ± 0.00	20.83 ± 6.29	22.08 ± 14.14	17.08 ± 0.00	12.08 ± 5.00	9.08 ± 7.58	3.75 ± 2.89
**4 kHz**	0 ± 10.76	18.75 ± 10.41	24.17 ± 12.50	25.42 ± 7.07	20.42 ± 0.00	18.42 ± 7.58	17.42 ± 13.51	10.42 ± 5.00
**8 kHz**	0 ± 13.05	24.58 ± 8.66	29.58 ± 10.00	29.58 ± 7.07	29.58 ± 0.00	27.58 ± 6.71	15.58 ± 9.62	12.92 ± 2.89
**16 kHz**	0 ± 7.72	22.08 ± 2.89	30.00 ± 4.79	26.25 ± 3.54	28.75 ± 0.00	28.75 ± 5.00	16.75 ± 9.75	12.08 ± 5.77
**32 kHz**	0 ± 9.64	18.75 ± 2.89	25.42 ± 7.07	27.92 ± 3.54	25.42 ± 0.00	20.42 ± 7.91	16.42 ± 9.62	15.42 ± 5.00

^1^ In Group VI, recordable auditory evoked potentials were obtained only in one rat.

## Data Availability

The data presented in this study are available in the article.

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
