# Peer review of "Efficacy and Mechanisms of Antioxidant Compounds and Combinations Thereof against Cisplatin-Induced Hearing Loss in a Rat Model"

_antioxidants, 2024, doi:10.3390/antiox13070761_

Round 1
Reviewer 1 Report
The study provides valuable insights into the otoprotective effects of antioxidant formulations against cisplatin-induced ototoxicity. The authors have selected a rat model to investigate the efficacy and mechanisms of antioxidant compounds against cisplatin-induced hearing loss.
The choice of a rat model has both advantages and disadvantages that should be addressed in the paper. While rats share many similarities with humans, there are also significant differences in cochlear anatomy, metabolism, and drug metabolism. Therefore, findings from rat studies may not always directly translate to humans. Rats may respond differently to cisplatin compared to humans, particularly in terms of the extent and pattern of ototoxic damage. This could affect the generalizability of the findings to clinical settings. Additionally, the authors could discuss how they have addressed the limitations and how their findings may contribute to our understanding of cisplatin-induced hearing loss and potential treatments in humans. The findings have important clinical implications for mitigating cisplatin-induced hearing loss in cancer patients, potentially improving their quality-of-life post-treatment. Further studies are needed to validate these findings in clinical settings and to explore the long-term effects of antioxidant supplementation on cisplatin ototoxicity.
The rationale for using a combination of antioxidant compounds is well-explained, but it could be clearer how each component contributes to the overall antioxidative effect. It might be beneficial to specify the rationale behind choosing the particular doses of each compound and the treatment duration, as well as reference any previous studies that informed these decisions. It's important to clarify whether the doses and combinations of otoprotective agents were chosen based on previous literature or preliminary experiments, and if any pilot studies were conducted to optimize the treatment protocols.
The use of a combination of vitamins A, C, E, and Mg++ (ACEMg) is intriguing and appears to show promising results in protecting against auditory damage caused by cisplatin. The finding that NAC degrades the otoprotective effects of ACEMg is unexpected and warrants further investigation into the underlying mechanisms. It's interesting to note the dose-dependent effect of NAC, where higher doses lead to a loss of otoprotection activity. This suggests a delicate balance in antioxidant dosing for optimal protection. The concept of an "antioxidation threshold" is thought-provoking and underscores the importance of carefully evaluating antioxidant combinations and doses to avoid diminishing returns or adverse effects.
Moreover, the observation that Mg++ alone can prevent cisplatin-induced ototoxicity is significant and raises questions about its mechanism of action and potential synergistic effects with other antioxidants.
The choice of initiating otoprotective treatments five days prior to the induction of cisplatin ototoxicity in the experimental protocol raises an important consideration. This timing allows for the antioxidants to accumulate and potentially exert their protective effects before the ototoxic insult occurs. However, the rationale behind this specific timing should be further justified. While it is common practice in animal studies to allow for a pre-treatment period to ensure maximum efficacy of the antioxidants, the justification for the five-day interval should be clarified. This could include discussing the pharmacokinetics and pharmacodynamics of the antioxidant compounds used and any previous literature supporting this timing. Also, translating this timing to human application poses challenges. In clinical settings, the timing of otoprotective treatments may need to be adjusted based on various factors such as the specific antioxidant compounds used, the treatment regimen, and the patient's medical history. Therefore, further discussion on how this protocol could be applied in a clinical context would be beneficial. Overall, while the choice of initiating otoprotective treatments prior to cisplatin administration is logical, further justification and discussion on the specific timing are needed.
The use of a sound-proof booth and anesthesia with isoflurane ensures accurate and controlled measurements of auditory thresholds in the animals. However, it's important to note that Group VI (NAC 500 + ACE + Mg) was excluded from the statistical analysis of threshold changes due to technical issues. While this exclusion is understandable, it's crucial to acknowledge any limitations in data collection and analysis. The inability to perform statistical analysis for Group VI-NAC500+ACEMg due to unstable ABR recordings indicates the need for cautious interpretation of its efficacy.
The choice of decalcifying solution (50% RDO) is appropriate for removing calcium deposits from the cochlear tissue, which can interfere with histological staining; please add this info in the text.
It would be beneficial to discuss the need for balancing otoprotective effects with antineoplastic efficacy.
Please move reference [86] earlier in the discussion and create a separate paragraph for the conclusions; this change would enhance the structure and clarity of the discussion section.
Author Response
Major comments
The study provides valuable insights into the otoprotective effects of antioxidant formulations against cisplatin-induced ototoxicity. The authors have selected a rat model to investigate the efficacy and mechanisms of antioxidant compounds against cisplatin-induced hearing loss.
Thank you very much for your thoughtful, constructive review. Your comments and criticisms have greatly helped to improve the quality and readability of the paper.
Detail comments
The choice of a rat model has both advantages and disadvantages that should be addressed in the paper. While rats share many similarities with humans, there are also significant differences in cochlear anatomy, metabolism, and drug metabolism. Therefore, findings from rat studies may not always directly translate to humans. Rats may respond differently to cisplatin compared to humans, particularly in terms of the extent and pattern of ototoxic damage. This could affect the generalizability of the findings to clinical settings. Additionally, the authors could discuss how they have addressed the limitations and how their findings may contribute to our understanding of cisplatin-induced hearing loss and potential treatments in humans. The findings have important clinical implications for mitigating cisplatin-induced hearing loss in cancer patients, potentially improving their quality-of-life post-treatment. Further studies are needed to validate these findings in clinical settings and to explore the long-term effects of antioxidant supplementation on cisplatin ototoxicity.
We have added a “study limitations” section at the end of the discussion, to clarify the important points that you have raised. See lines 646-670 in the revised manuscript.
The rationale for using a combination of antioxidant compounds is well-explained, but it could be clearer how each component contributes to the overall antioxidative effect.
The last paragraph of the Introduction deals with the mechanism contributed by each compound to antioxidation. Following your suggestion, we have split it in three paragraphs, adding more information to make this point clearer, as you suggest. Please see lines 125-163 in the revised version of the manuscript.
It might be beneficial to specify the rationale behind choosing the particular doses of each compound and the treatment duration, as well as reference any previous studies that informed these decisions. It's important to clarify whether the doses and combinations of otoprotective agents were chosen based on previous literature or preliminary experiments, and if any pilot studies were conducted to optimize the treatment protocols.
Doses and treatment duration were based on previously published studies, as specified in Materials and Methods, under section 2.2. Additional information has now been now added to clarify this point. See lines 199-204 in the revised version of the paper.
The use of a combination of vitamins A, C, E, and Mg++ (ACEMg) is intriguing and appears to show promising results in protecting against auditory damage caused by cisplatin. The finding that NAC degrades the otoprotective effects of ACEMg is unexpected and warrants further investigation into the underlying mechanisms. It's interesting to note the dose-dependent effect of NAC, where higher doses lead to a loss of otoprotection activity. This suggests a delicate balance in antioxidant dosing for optimal protection. The concept of an "antioxidation threshold" is thought-provoking and underscores the importance of carefully evaluating antioxidant combinations and doses to avoid diminishing returns or adverse effects.
Thanks for the insightful comments and suggestions. Indeed, the reported results intend to be, among other things, a “call of attention” about possible consequences of “too much” antioxidation. Our results support that that may be the case for the NAC500+ACEMg combination. The “paradoxical” limitation of antioxidant activity may not be so “paradoxical” when the notion of antioxidation threshold or window is incorporated.
Moreover, the observation that Mg++ alone can prevent cisplatin-induced ototoxicity is significant and raises questions about its mechanism of action and potential synergistic effects with other antioxidants.
The role of Mg++ is considered in detail in the discussion. See lines 539-548 in the revised manuscript.
The choice of initiating otoprotective treatments five days prior to the induction of cisplatin ototoxicity in the experimental protocol raises an important consideration. This timing allows for the antioxidants to accumulate and potentially exert their protective effects before the ototoxic insult occurs. However, the rationale behind this specific timing should be further justified. While it is common practice in animal studies to allow for a pre-treatment period to ensure maximum efficacy of the antioxidants, the justification for the five-day interval should be clarified. This could include discussing the pharmacokinetics and pharmacodynamics of the antioxidant compounds used and any previous literature supporting this timing. Also, translating this timing to human application poses challenges. In clinical settings, the timing of otoprotective treatments may need to be adjusted based on various factors such as the specific antioxidant compounds used, the treatment regimen, and the patient's medical history. Therefore, further discussion on how this protocol could be applied in a clinical context would be beneficial. Overall, while the choice of initiating otoprotective treatments prior to cisplatin administration is logical, further justification and discussion on the specific timing are needed.
Although, as it is well known, there is great deal of empirical approach to the timing of experimental preventative antioxidant treatments, there seems to be an “unwritten consensus” in that treatments initiated 5 to 7 days prior to the ototoxic exposure seem to provide a safe window to unravel mechanisms. However, as you point out, this can probably not be extended to therapeutic windows in potential applications in the human clinic. Among other things, the injection route used in this study and others, although adequate for a proof-of-concept stage, is unlikely to be of therapeutic use in humans because of possible interference with the antineoplastic mechanisms of cisplatin, which involves oxidative stress in tumoral cells. Following your advice, we have clarified these issues by adding sentences in the materials and methods and discussion section, including the limitations of the study. See lines 199-204 and 664-670 in the revised manuscript.
The use of a sound-proof booth and anesthesia with isoflurane ensures accurate and controlled measurements of auditory thresholds in the animals. However, it's important to note that Group VI (NAC 500 + ACE + Mg) was excluded from the statistical analysis of threshold changes due to technical issues. While this exclusion is understandable, it's crucial to acknowledge any limitations in data collection and analysis. The inability to perform statistical analysis for Group VI-NAC500+ACEMg due to unstable ABR recordings indicates the need for cautious interpretation of its efficacy.
The use of the term “unstable” is misleading, and we apologize for that. In the NAC500+ACE+Mg treatment group only one animal had recordable activity. The rest of animals in this group did not show recordable activity. Indeed, this reinforces the notion that extensive antioxidant combinations may lead to loss of antioxidant activity. The only animal showing detectable thresholds should be seen as an outlier, likely a product of natural variability which we chose to report and not eliminate from the experimental series, to strengthen a main point of the results. We have further clarified this issue in the results (see lines 352-355) and in the discussion (see lines 568-577) sections of the revised manuscript.
The choice of decalcifying solution (50% RDO) is appropriate for removing calcium deposits from the cochlear tissue, which can interfere with histological staining; please add this info in the text.
We have added a statement to clarify that decalcification time in RDO was adjusted in pilot assays so that it did not interfere with staining. See lines 249-251 in the paper.
It would be beneficial to discuss the need for balancing otoprotective effects with antineoplastic efficacy.
A comment on this issue has been added in the discussion. See lines 663-670 in the revised version of the manuscript.
Please move reference [86] earlier in the discussion and create a separate paragraph for the conclusions; this change would enhance the structure and clarity of the discussion section.
We agree with you, and we have proceeded accordingly. A separate paragraph for the conclusions has been created. See lines 672-680 in the revised paper.
Reviewer 2 Report
Authors insisted that antioxidant formulations, specifically vitamins A, C, E, and Mg++ (ACEMg), protect against this side effect in rats. In addition, authors considered that combining ACEMg with N-acetylcysteine (NAC) degrades its protective effect, highlighting the importance of balanced antioxidant dosing. This study is interesting, but there are several concerns to publish in current version.
Major concerns
1. The author has shown that oxidative stress induced by cisplatin and the administration of drugs against it prevent the onset of hearing loss, but there are discrepancies between Figure 1 and Figure 4. For example, Group V showed hearing loss in Figure 1, but in Figure 4, oxidative stress was reduced, which was inconsistent.
2. Groups VII, VIII, and IX also appear to have differences in staining. Quantitative evaluation such as Western blot or ELISA is needed.
3. The concentration of ACEMg in the body and in the cochlea may differ depending on the substance, although it was administered 5 days before and until the second day of cisplatin administration.
4. The study also states that Mg causes diarrhea in humans, but it should be stated whether there were no side effects in this animal study. If not, we believe that if Mg is used at that concentration in humans, there will be no side effects.
Minor concerns
1. How do you explain the fact that no OHCs were damaged in Figure 2?
View Synonyms and Definitions
Author Response
Major comments
Authors insisted that antioxidant formulations, specifically vitamins A, C, E, and Mg++ (ACEMg), protect against this side effect in rats. In addition, authors considered that combining ACEMg with N-acetylcysteine (NAC) degrades its protective effect, highlighting the importance of balanced antioxidant dosing. This study is interesting, but there are several concerns to publish in current version.
Thanks for your positive comment and constructive review. We believe that we have addressed your criticisms and concerns. Please see specific responses below.
Major concerns
- The author has shown that oxidative stress induced by cisplatin and the administration of drugs against it prevent the onset of hearing loss, but there are discrepancies between Figure 1 and Figure 4. For example, Group V showed hearing loss in Figure 1, but in Figure 4, oxidative stress was reduced, which was inconsistent.
Thanks for your comment. Actually, we mention openly this issue in the discussion. Group V-NAC500+Mg poses indeed an interpretation challenge, because it is the only treatment group in which a comparatively poorer threshold preservation is not mirrored by higher 3-NT immunolabeling and larger loss of OHCs. At this stage, we can only propose possibilities open to experimental testing. For instance, it could be that apparently “healthy” OHCs, in which relative 3-NT immunolabeling was measured, do not provide sufficient amplification, thus increasing thresholds, if the stria vascularis is still subject to oxidative stress, after NAC500+Mg. This could happen, for example, if NAC concentration in the stria vascularis is increased to toxic levels due to excess availability mediated by Mg vasodilation of the capillary bed of the stria vascularis. The stria vascularis seems to be particularly sensitive to cisplatin-mediated oxidative stress (see Gibaja et al., 2022). We have expanded the part of the discussion section dealing with this issue, by adding information at the end of the last paragraph of section 4.2. See lines 614-644 in the revised version of the manuscript.
- Groups VII, VIII, and IX also appear to have differences in staining. Quantitative evaluation such as Western blot or ELISA is needed.
Thanks for your comment and suggestion. 3-NT staining falls within what is expected for treatment groups VII, VIII and IX. Actually, as shown in the bar graph of Figure 4, they are similarly low, close to control levels.
The choice of relative quantification of 3-NT immunolabeling should be seen in the global context of the methods utilized in the study. We have added a few sentences highlighting the rationale for this choice. See lines 614-619 in the revised version of the manuscript.
Although we fully agree that the resources of Western blot or ELISA provide important information on overall protein levels, we chose 3NT immunohistology as a valid tool to evaluate oxidative stress from a spatial perspective. Introducing protein analysis in this study would have resulted in a very large number of animals, in the context of an already animal-intensive experimental design and analysis.
- The concentration of ACEMg in the body and in the cochlea may differ depending on the substance, although it was administered 5 days before and until the second day of cisplatin administration.
We agree with this comment. Actually, it is a variable which is not possible to control at this stage. Cochlear concentrations would require unavailable detection methods and measuring Mg++ in plasma is unreliable because of its fast incorporation into cells. This limitation is acknowledged in section 4.3 of the discussion.
- The study also states that Mg causes diarrhea in humans, but it should be stated whether there were no side effects in this animal study. If not, we believe that if Mg is used at that concentration in humans, there will be no side effects.
There was no diarrhea in our experimental series. A statement has been added to clarify this issue. See lines 547-548 in the revised manuscript.
Minor concerns
- How do you explain the fact that no OHCs were damaged in Figure 2?
Thanks for pointing this out. We noticed the presence of some individual cases in several treatment groups in which there was little or no loss of OHCs. It is interesting that these outliers belong mostly into treatment groups providing best antioxidant otoprotection. They may represent cases of exceptional sensitivity to antioxidant otoprotection, in the context of natural biological variability or alternatively, limited sensitivity to cisplatin ototoxicity. It is worth noting that neither the Cis group, nor NAC-1000 or NAC500+ACE+Mg show such individual outliers. We show individual OHC counts alongside the average count (black graph lines in Fig. 2) to give an idea of this intrinsic variability. We have added an explanation in the legend for figure 2. See lines 411 to 416 in the revised version of the manuscript.
Round 2
Reviewer 1 Report
The authors addressed all the issue I raised.
-
Reviewer 2 Report
All concerns are appropriately corrected.
No problem.